# Testifying after an Investigation: Shaping the Mental Health of Public Safety Personnel

**DOI:** 10.3390/ijerph192013643

**Published:** 2022-10-21

**Authors:** Rosemary Ricciardelli, R. Nicholas Carleton, Barbara Anschuetz, Sylvio Gravel, Brad McKay

**Affiliations:** 1Fisheries and Marine Institute, Memorial University of Newfoundland, St. John’s, NL A1C 5S7, Canada; 2Psychology Department, University of Regina, Regina, SK S4S 0A2, Canada; 3The Trauma Centre, Cape Town 7925, South Africa; 4Badge of Life Canada, Orillia, ON L3V 5X6, Canada

**Keywords:** potentially psychologically traumatic event, public safety personnel, testifying, investigation

## Abstract

In this editorial, we draw on two Canadian cases to interrogate how mass causality events and investigations consume many responders before (e.g., public safety communicators, detachment service assistants), during (e.g., police, fire, paramedics), and after the incident (e.g., coroners, correctional workers, media coverage). Their well-being may suffer from the associated processes and outcomes. In the current article, we focus on the mass causality incident of 2020 in Nova Scotia, Canada, and the investigation following a prisoner death in 2019 in Newfoundland, Canada, to explore how testifying post-incident can be made more palatable for participating public safety personnel (PSP). Specifically, we study how testifying after an adverse event can affect PSP (e.g., recalling, vicarious trauma, triggers) and how best to mitigate the impact of testimony on PSP well-being, with a lens to psychological “recovery” or wellness. We focus here on how to support those who may have to testify in a judicial proceeding or official inquiry, given being investigated for best-intended actions can result in moral injury or a posttraumatic stress injury, both exacerbated by judicial review, charge, accusation, or inquiry.

## 1. Introduction

Examples of potentially psychologically traumatic events (“PPTEs”) include, but are not limited to, adverse childhood experiences, exposure to war as a combatant or civilian, threatened or actual physical assault, threatened or actual sexual violence, being kidnapped, being taken hostage, torture, natural or human-made disasters, or severe physical injuries such as motor vehicle events and industrial accidents [1,2]. Exposure to a PPTE can be direct, witnessed, learned about, and can occur to a close family member or close friend (in such cases the event must have been violent or unintentional), or through repeated or extreme exposures to aversive details [1,2]. A mass causality event, or any investigation after an incident for public safety personnel (PSP), constitutes a PPTE. PSP [1] include diverse professionals (e.g., border services personnel, correctional workers, fire Marshall investigators, coroners, firefighters, operational and intelligence personnel, paramedics, police, public safety communicators, search and rescue personnel). PPTEs can cause psychological trauma that may be consistent with one or more posttraumatic stress injuries (PTSIs), including but not limited to posttraumatic stress disorder (PTSD), major depressive disorder (MDD), panic disorder, and generalized anxiety disorder (GAD) [1,2,3]. The phrase “psychologically traumatic event” is often preceded by the word “potentially” to underscore the importance of dynamic individual and environmental contextual factors that influence whether an event was, or is perceived as, psychologically traumatic for any given individual at any given time [1,2,3]. Relevant to the current editorial, we reflect on two Canadian cases to show how mass causality events and investigations consume many responders before (e.g., public safety communicators, detachment service assistants), during (e.g., police, fire, paramedics) and after the incident (e.g., coroners, correctional workers, media coverage), whose well-being may suffer from the associated processes and outcomes. Thus, we draw on the mass causality incident of 2020 in Nova Scotia, Canada, and the investigation following a prisoner death in 2019 in Newfoundland, Canada, to describe incidents that may be compromising to PSP wellness. We then explain the relationship between investigation and testimony and PSP mental health, recognizing that being investigated for best-intended actions can result in moral injury or a PTSI and that both are exacerbated by judicial review, charge, accusation, or inquiry. Indeed, court proceedings are challenging for PSP, victims, civilian witnesses, jurors, court staff, and counsel. We then speak to how testifying post incident, for instance in a judicial proceeding or official inquiry, can be made more palatable for participating PSP, recognizing the impacts of recalling PPTE, vicarious trauma, triggers, etc. Herman (2003), a recognized authority on trauma, has stated, “Indeed, if one set out intentionally to design a system for provoking symptoms of posttraumatic disorder, it might look very much like a court of law” [4] (p. 159).

## 2. Two Cases

The first case we consider here is the Mass Causality event in rural Nova Scotia that occurred from April 18 to 19, 2020. To summarize, between 10pm on 18 April and 5am on 19 April, 13 individuals were killed and another shot at by a perpetrator (name omitted to reduce publicity) [5]. PSP responded to the calls for service, the first being a woman reporting the shooting of her husband and calls continued until Royal Canadian Mounted Police Officers (RCMP) apprehended and fatally wounded the perpetrator. The event involved diverse first responders and PSP, resulting in extensive media coverage, given it was the “worst mass shooting in Canadian history” [6]. Following the incident and investigation, a Mass Casualty Commission remains in operation [5]. In this commission, of which two authors have participated, PSP who participated in the events have been called to testify, cross-examined, and discussion continue with the goal being “to provide meaningful recommendations to help make communities safer in the future” [5].

The second case we consider here is the in-custody death of Jonathan Henoche. On the 6 November 2019, Henoche, involved in an incident with correctional officers, became unresponsive and died [7]. In total, 10 officers faced criminal charges—criminal negligence causing death, manslaughter, and manslaughter and failure to provide necessities of life—, with charges eventually dropped against all officers, but largely after a grueling public process with slandering media coverage and years of time on forced leave [8]. The ten officers now “claim they suffered due to negligence of police, prosecutors and the chief medical examiner” [9] and, in response, have initiated a lawsuit again the provincial government for their alleged malicious prosecution where they experienced “mental anguish, loss of reputation and finances, and a sense of betrayal” [9].

The two cases above have in common both that they involved PSP, who fulfilled their occupational responsibilities to the best of abilities during a PPTE exposure and were subsequently investigated and on trial, where they had to relive a PPTE for the purposes of “justice”, while also trying to resolve the impacts of the PPTE and subsequent events on self.

## 3. PPTE, PTSD, and Mass Causalities and Investigations

Not everyone who is exposed to a PPTE develops a PTSI [1,2]. A PTSI describes a range of problems including, but not limited to, mental disorders such as PTSD and mental health conditions that may not meet Diagnostic and Statistical Manual of Mental Disorders (DSM) or International Classification of Diseases (ICD) criteria for PTSD but still interfere with daily functioning in social, work, or family activities [1]. Pre-existing factors, concurrent, or post-PPTE mental and physical injuries can all impact risk for a PTSI [2]. Examples include prior history of unresolved PPTE exposures, perceived helplessness during the PPTE, perceived uncertainty during the PPTE, and perceived social support post-PPTE [2].

A PPTE can be further complicated by acts of omission or commission, betrayal, amoral or immoral leadership, or organizational failures, which can cause complicating factors such as moral injury [10], referring to incidents that violate the morals, ethics, and values of an individual [1]. Individual differences in experiences and environment across PSP may serve as risk factors for psychopathology; for example, previous experiences of PPTE and other stressors (e.g., adverse childhood events, life stressors) [11,12,13,14], pre-existing psychopathology [15,16], rumination [16], peritraumatic distress (i.e., the emotional and physiological distress experienced during and/or immediately after a PPTE) [17,18], and peritraumatic uncertainty (i.e., the uncertainty and distress experienced during and/or immediately after a PPTE) [19,20,21]. There are also individual differences in psychological variables that may serve as risk factors for PTSI [22]; for example, world view, social support, and social adjustment [18,23,24,25], and maladaptive self-appraisals (i.e., how you see yourself, the world or your symptoms in the aftermath of trauma as well as self-blame or other trauma attributions) [26,27]. There are also individual differences that may serve as risk factors for psychopathology; for example, perceived interpersonal support (e.g., positive relationships with friends, spouse, family, colleagues) [25,28,29], and positive life activities (e.g., hobbies, exercise) [29,30,31].

Relevant to the mass casualty, case one, the RCMP were the front line. In general, the RCMP report the highest average number of exposures to PPTE relative to other Canadian PSP, often reporting more than 11 exposures to each different type of PPTE [3]. RCMP officers have reported among the highest proportion of positive screenings for PTSD (30.0%) and MDD (31.7%) among Canadian PSP, with half (50.2%) of the RCMP screening positive for one or more mental health disorders at any given time [32]. Substantial percentages of RCMP have reported past-year and lifetime suicidal behaviors (i.e., ideation [9.9%, 25.7%, respectively], planning [4.1%, 11.2%, respectively], attempts [0.2%, 4.2%, respectively]) at rates approximately double that of the general population [33]. RCMP officers report more suicidal ideation than other police officers in Canada and score higher on measures of stress, PTSD, MDD, GAD, and panic disorder [34]. More than half of RCMP (55.4%) report comorbid mental health challenges [35] and difficulties with insomnia (59%) [36]. The RCMP also report substantial organizational and operational stressors relative to other Canadian PSP organizations [37], as well as the lowest levels of much-needed social support [25,38,39]. Families of police officers also experience unique stressors related to service [40,41], which can be exacerbated when the officer experiences one or more PTSI [37,41].

Relevant to the officers impacted by the in-custody death, case two, correctional officers were the first responders. In general, for correctional workers, they too report among the highest average number of PPTE exposures in comparison to other PSP [3,42]. Correctional workers report positive screens for PTSD at a prevalence of 29.1%, 31.1% for MDD, and 54.6% screen positive for at least one mental health disorder [32]. Among provincial correctional workers in the province of Ontario, 58.2% screen positive for at least one mental health disorder [43]. The number of correctional workers who report past-year and lifetime suicide behaviors is high (i.e., ideation [11%, 35.27%, respectively], planning [4.8%, 20.1%, respectively], attempts [0.4%, 8.1%, respectively]) [33]. Among provincial correctional workers in the province of Ontario, the prevalence is also quite high (i.e., ideation [7.0%, 26.6%, respectively], planning [2.6%, 11.9%, respectively], attempts [redacted, 5.2%]) [44]. Correctional workers also report comorbid mental health challenges [35] and difficulties with insomnia (58%) [36]. They experience organizational and operational stressors [37], as well as low levels of social support [25,38].

All PSP who responded to the mass casualty event or who were involved in the death in-custody would have necessarily been directly or indirectly exposed to a complex, intense, and extraordinary series of interactive PPTEs. These PPTEs would involve people with whom they have personal or professional relationships, and experienced protracted peritraumatic distress over as many as 14 h of exposure to the diverse PPTEs. In addition, PSP likely entered into the mass casualty event or in-custody death with a history of frequent PPTE exposures; were at risk for frequent and extraordinary PPTE exposures after the active shooter call or death, assuming PSP continued to serve; and were at an increased risk for having pre-existing mental health disorders as a result of previous PSP experiences. The PSP were also at an increased risk of having experienced diverse and protracted organizational and operational stressors before and after the incident; compromised social supports before and after the PPTEs; and an increased risk for rumination, sleep difficulties, and maladaptive self-appraisals related to the experience. Simply said, extant research suggests the mass casualty event—as well as the Newfoundland investigation— would qualify as a complex, intense, and extraordinary PPTE [2,3,10,42].

In addition, the involved PSP who responded to the event in any way would have been, and would continue to be, at increased risk for experiencing the event as psychologically traumatic. Per the definition of PPTE, the ultimate decision for whether the event was perceived as psychologically traumatic, or is currently perceived as psychologically traumatic, necessarily rests with each individual PSP. There is also reason to believe the current perceptions each PSP hold of the event may change over time because of a great number of internal and external resiliency and risk factors, including but not limited to personal and professional supports, or exposures to additional stressors including but not limited to additional PPTE.

The probability is that PSP involved in the mass casualty event or in-custody death would have experienced protracted peritraumatic distress during and immediately afterwards. Most humans are generally resilient to stressors, including PPTE exposures [2]; however, the repeated PPTE exposures experienced by PSP [3], coupled with other occupational stressors [45] and reduced social supports [25,38] would have increased the risk for several PTSI among PSP exposed to the event. The protracted nature of the subsequent media coverage, the investigation, and vocational requirements to remain engaged with the event would have further increased the risk for PTSI as a function of forced rumination [16], uncertainty [19,20,21], potentiated moral injuries [10], and maladaptive self-appraisals [26].

Anecdotal and experiential knowledge collected by the authors suggests there are other challenges that can and often do result from PPTE exposures for PSP. The challenges include denial, wherein PSP who are suffering try to ignore difficulties resulting from a PPTE experience. The PSP often experience stigma that inhibits their ability to admit a PPTE has had a negative impact [46]. Other evidenced outcomes include loneliness, inherent in feeling misunderstood as PSP progress through an investigation and an inquiry. Moreover, differing from slower operations, speed is a critical factor during a mass causality event or a critical incident and can eliminate time for consultation. PSP in such situations may be alone when making decisions, unable to draw on collegial expertise and experience. The same PSP may then face scrutiny from the public, their employer, and others. The experiences of PSP after a mass casualty event or critical incident facing investigation remain under researched, but ethnographic data provides important initial information. The sequalae, including the scrutiny, can result in many different debilitating outcomes, from second guessing oneself and losing self-confidence to loss of emotional control. Physical impacts also occur, such as loss of sleep and appetite control [30] as do social and psychological (e.g., reactions to public perception and media coverage; author forthcoming). The pervasive stress may also be tied to fears of losing peer support during the investigative process, which can be exacerbated by outcomes inherent to the process (e.g., lack of self-confidence), or fears of being unable to protect their family and loved ones from negative outcomes (e.g., emotional distress) driven by different sources (e.g., media coverage).

## 4. Barriers to Treatment

Acknowledging the potential for PSP involved in investigations and testimonies to experience PTSIs, it is relevant that PSP appear to face diverse hurdles to accessing mental health supports from their organization [45,47,48,49]. This means that PSP who may be of compromised well-being may not seek treatment. One explanation is that of stigma, which appears to be a distinct barrier PSP experience with respect to help-seeking behavior for mental health [50,51,52,53]. Self-stigma (e.g., internalized shame) and public stigma (e.g., external derogatory feedback) have both been inversely associated with help-seeking [54,55]. Mental health stigma may also influence mental health symptom reporting, hinder realistic prevalence estimates, limit overall resource availability [48,56,57], and create cycles that impede help-seeking. PSP have often reported perceiving symptoms of a mental disorder as associated with one being incapable, incompetent, weak, or a failure [58,59].

PSP also report substantial internal and external pressure to maintain a strong persona, which may compromise self-awareness regarding symptoms [50,60] and inhibit accessing professional services for fear of retribution by peers or administration [47,48,51,52,61]. PSP who the meet criteria for PTSD may endorse higher levels of mental health stigma [53]. The often-stoic environment of public safety professions appears problematic for all involved, particularly women, who appear to experience exacerbated stigma related to mental health [62].

The available evidence suggests PSP experiencing mental health challenges are most likely to turn to spouses or family for support, and least likely to turn to leaders or professionals [63]; however, PSP may feel uncomfortable disclosing mental health struggles to anyone, further reducing social support and help-seeking behavior [47,48]. There have been and continue to be growing efforts to reduce mental health stigma among Canadian PSP to improve self-awareness and increase help-seeking [40,64].

## 5. Ways Forward

PSP testifying after a PPTE, particularly a mass casualty event or investigation, may be experiencing PTSI symptoms and require considerations, supports, and in some cases accommodations. Ideally, PSP participation should be predicated on a meaningful rationale for their role as a witness, should be minimally invasive, and occur for a minimal number of times. Moreover, PSP may require peer support and professional mental health support from persons external to the stakeholders and organizations involved. PSP need to have agency over their participation and some sense of control to support their psychological health. A trauma-informed approach might include flexibility with testifying options, such as considering providing PSP with options for asynchronous testimony (e.g., accept written statements of response to specific questions or testifying live but in private to reduce the impact of the setting). Testimony by a PSP can require “re-living” the event, and for PSP experiencing a PTSI, the process of cross-examination by lawyers can be particularly challenging.

There are also organizational factors that may assist in easing the procedural impact of testifying and investigation. Central here is communication. Wherever possible, there needs to be a communication strategy that involves all stakeholders in an incident. All information that will not jeopardize the investigation or court proceedings should be transparently shared with PSP to provide them understanding of procedures and to provide support during the event; knowledge, in this sense can help PSP feel some semblance of control and agency while fulfilling their legal and occupational responsibilities. In addition, there is a need to prepare PSP for the court environment with the support of professional clinicians, who can help counter negative thoughts and the re-living of the PPTE. Understanding and mitigating harmful impacts of testimony, investigation, and PPTE exposures require more directed investigation and empirical assessment. However, given that investigations are commonplace for PSP, there is an urgent need for safeguards to support PSP while the processes continue. Safeguards may include organizational commitments of support for the PSP and their families after a PPTE, that occur before, during, and after any investigation and testimony. Peer support and professional mental health support for PSP proceeding through the process may be critical for the mental wellness of the PSP and their families.

Our work here is limited, given we are basing our analysis on experientially acquired knowledge, both from testifying and from lived experiences. We draw on informal and formal conversation rather than an empirical study, and recognize that such a study is needed in the future. However, more efforts must be made to protect those who protect citizens, particularly in the aftermath of PPTE.

## 6. Conclusions

Overall, our purpose in the current article was to operationalize investigation and trauma using Canadian cases and then to explore the relationship between mental health and experiences of investigation and testimony for PSP. We conclude by speaking to how PSP can be better supported during such processes to positively inform their mental health and well-being. Testimony is challenging, mentally draining, and difficult (with many collateral consequences), thus, it behooves society to ensure PSP are supported during such processes and efforts are made to make these experiences more tenable.

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
