# Peer review of "Testifying after an Investigation: Shaping the Mental Health of Public Safety Personnel"

_ijerph, 2022, doi:10.3390/ijerph192013643_

Round 1
Reviewer 1 Report
See attached.

Author Response
Reviewer 1:
I was not clear what the purpose of this manuscript is in terms of it being a narrative review, a case study, or something else as the methodology is not provided?? For a narrative review for example PRISMA guidelines can be followed. For case studies, one would have to have more information about those involved with the cases and what their levels of PPTE-related trauma is, what they experienced, etc. It seems that what is written in what could have transpired with these individuals but there are no data to support this.
Response: The article is an editorial case study based on experiences and insights gained through experiences of testifying and ethnographic observations around other PSP testifying. We now make this clear in the article.
While the topic is important, there needs to be an emphasis on testifying. The title of the manuscript is testifying however that gets lost throughout the paper. Also, there is much redundancy throughout as sentences and topics are repeated over and over. Also, when multiple data points are reported, the term respectively should be used to separate which data points are being discussed at that time. For example, see page 3 lines 140-143 where there are multiple percentages in brackets referring to past-year and lifetime time periods-the term respectively would allow one to make sure he or she is looking at the right time period for the date (e.g., 11%, 35.2%, respectively).
Response: We have clarified the emphasis on testifying so it is centralized in the article. We have tried to reduce and remove redundancy. We incorporated the term respectively when presenting multiple data points.
On page 6, there is discussion that this manuscript provides anecdotal and experiential knowledge of the authors however to be published as an article I would recommend a more formal methodology be provided as stated above and conducted. I see this manuscript as presented now more of a perspective or editorial rather than research-based per se.
There are some grammatical errors throughout such as conditions being capitalized when they should not be, especially when provided with a capitalized acronym. I am sure the editors will be able to correct these types of concerns (and others) with the authors.
Response: We have removed grammatical challenges and errors. We now present the article as a case study/ editorial.
Reviewer 2 Report
Dear Author (s),
Although the topic is very interesting, the methodology is not strong enough to ensure the generalizability. I suggest you to work further on the methodology and findings. Is it possible to explain more than 2 cases (you used 2 cases)? If not, then what about using other type of data (primary or secondary). Overall, the methodology and findings should be completely re-designed to improve the quality of the manuscript. Good Luck!!!
Author Response
Reviewer 2:
Although the topic is very interesting, the methodology is not strong enough to ensure the generalizability. I suggest you to work further on the methodology and findings. Is it possible to explain more than 2 cases (you used 2 cases)? If not, then what about using other type of data (primary or secondary). Overall, the methodology and findings should be completely re-designed to improve the quality of the manuscript. Good Luck!!!
Response: Thank you. We have revised to ensure the article is an editorial / case study.
Round 2
Reviewer 1 Report
See attached.

Author Response
Second Review of “Testifying after an investigation: Shaping the mental health of public safety personnel
The revision of this manuscript is improved but there are still many areas that need to be addressed. First, I would remove from the title the words before the colon. The information about testifying has been added but it is only at the end. The title could be something such as Improving the Mental Health of Public Safety Personnel to Prevent Negative Outcomes or something like that. I leave that to the authors and editors.
Response: We have revised the title.
On page one, paragraph 1, please remove the word accidental or accidents and use motor vehicle events (line 34) and violent or unintentional (line 36).
Response: We have revised accordingly.
There are still several grammatical issues in the paper so I would highly recommend an English expert review the paper for redundancies, run-on sentences, incorrect punctuations, etc. For example, in the first sentence under Introduction, potentially psychologically traumatic events should not be capitalized. Neither should medical disorders as listed on the bottom of page 1 and top of page 2 and throughout the manuscript. The acronyms are capitalized but not the conditions. On page 4 from line 166-182 the material is very redundant. Also, I could not find references 50-55, 57 or 76 in the text.
Response: We have reviewed for grammar, and made all suggested revisions.
Finally, I see this manuscript as perhaps a commentary or letter to the editor. It is still not clear what the role of the authors were in the cases and how they might have been affected. The court proceedings are not fully described but more the potential effects of such proceedings are. I am still somewhat confused over the direction of the manuscript in terms of is being a description of the cases, a need to call attention to this issue, directions for change, etc. All of this should be spelled out somewhere in the Introduction.
Response: We spell out in the introduction that the purpose of the paper is to draw attention to the challenges of investigation and testimony and that we put the case forward to better support PSP through such processes.
Reviewer 2 Report
To be honest, I can see minor changes in the revised version. If it is editorial or something similar (not a full research article) then the editor in charge may decide about the final acceptance. I suggest you to elaborate the discussion to ensure the contribution of your research. At present, it can't ensure the quality of a good manuscript.
Thank you.
Author Response
To be honest, I can see minor changes in the revised version. If it is editorial or something similar (not a full research article) then the editor in charge may decide about the final acceptance. I suggest you to elaborate the discussion to ensure the contribution of your research. At present, it can't ensure the quality of a good manuscript.
Response: We have elaborated the purpose of the article in the discussion.